

# Anthropogenic and natural drivers of a strong winter urban heat island in a typical Arctic city

Mikhail Varentsov[1,2], Pavel Konstantinov[1,3], Alexander Baklanov[4], Igor Esau[5], Victoria Miles[5] and Richard Davy[5]

[1] Lomonosov Moscow State University, Faculty of Geography / Research Computing Center, GSP-1, Leninskiye gory, 19991, Moscow, Russia
[2] A.M. Obukhov Institute of Atmospheric Physic of the Russian Academy of Science, Pyzhevsky pereulok, 3, 119017, Moscow, Russia
[3] Kola Science Centre of the Russian Academy of Sciences, Fersman str. 14, 184209, Apatity, Russia
[4] World Meteorological Organization (WMO), Avenue de la Paix, 7 bis, 1211, Genève, Switzerland
[5] Nansen Environmental and Remote Sensing Centre/Bjerknes Centre for Climate Research, Thormøhlensgate 47, 5006, Bergen, Norway

*Correspondence to*: Mikhail Varentsov (mvar91@gmail.com)

**Abstract.** The Arctic has rapidly urbanized in recent decades with two million people currently living in more than a hundred cities north of 65° N. These cities have a harsh but sensitive climate and warming here is the principle driver of destructive thawing, water leakages, air pollution, and other detrimental environmental impacts. This study reports on the urban temperature anomaly in a typical Arctic city. This persistent warm anomaly reaches up to 11 K in winter with the wintertime mean urban temperature being on average 1.9K higher in the city centre than in the surrounding natural landscape. An urban temperature anomaly, also known as an urban heat island (UHI), was found in remote sensing and in situ temperature data. High-resolution (1 km) model experiments run with and without an urban surface parametrization helped to identify the leading physical and geographical factors supporting a strong temperature anomaly in a cold climate. The statistical analysis and modelling suggest that direct anthropogenic heating contributes at least 50% to the observed UHI intensity, and the rest is created by natural microclimatic variability over the undulating relief of the area. The current UHI effect can be as large as the projected, and already amplified, warming for the region in the 21st century. In contrast to earlier reports, this study found that the wintertime UHI in the Arctic should be largely attributed to direct anthropogenic heating. This is a strong argument in support of energy efficiency measures, urban climate change mitigation policy, and against high-density urban development in polar settlements. The complex pattern of thermal conditions, as revealed in this study, challenges urban planners to account for the observed micro-climatic diversity in perspective sustainable development solutions.





## 1 Introduction

Rapid urbanization of the Arctic in recent decades (NORDREGIO, 2011) has strongly modified social (Parente et al., 2012) and environmental (Yu et al., 2015) conditions in the region. At present, there are more than a hundred urban settlements north of 65°N totalling around two million residents. The largest are the Russian cities of Murmansk (303 000 inhabitants), Norilsk and Novy Urengoy (both with more than 100 000 inhabitants). The more typical Arctic city has a population between 30 and 70 thousand people who are predominantly working in the mining industry. Although jobs in these cities are generally well-payed and the population is highly skilled, the cities are vulnerable to the known "extractivism" setback (Wilson et al., 2016) whereby their management is driven by external economic markets and socio-political agendas, and urban planning employs construction standards taken directly from those used in the temperate mid-latitudes (Shubenkov and Blagodeteleva, 2015; Parente et al., 2012). Importantly, urban planning does not account for the formation of the urban heat island (UHI) – an area of persistent warm anomalies of surface and air temperatures within cities.

The UHI phenomenon is frequently considered as a detrimental local climate effect inducing excessive thermal stress on humans and ecosystems in mid- and low-latitude cities (see e.g. Tan et al, 2010; Konstantinov et al, 2014; Wouters et al, 2017; EPA, 2018). The UHI impact can be particularly severe in megacities during summer heat waves. However, the winter UHI has not been considered as a problem in climate literature. Studies of mid-latitude UHIs have concluded that the urban temperature anomalies are not as strong in the winter months when there is less solar radiation and calm weather events are less frequent (e.g. Wilby, 2003; Gedzelman et al., 2003; Zhou et al., 2013; Choi et al., 2014; Kuznetsova et al., 2017).

However, the winter UHI has been found in all Arctic cities which have been studied, e.g. in Fairbanks (Magee et al, 1999), Norilsk (Varentsov et al, 2014), in all 28 cities in North-West Siberia (Esau and Miles, 2016; Miles and Esau, 2017; Konstantinov et al., 2018), as well as in Apatity (Konstantinov at al., 2015). Apatity is a typical, medium-size Arctic city that has been densely built-up with centrally heated apartment and public buildings of 5 to 9 storeys. Earlier analysis for Apatity, based on a short observation campaign, has reported that in winter the surface air temperature could be up to 5-10 K higher in the city than in the surrounding countryside (Konstantinov et al., 2015), which is surprisingly high for a medium-size city.

Higher winter temperatures in Arctic cities may induce both beneficial and detrimental socio-economic effects. Higher urban temperatures decrease health and infrastructure damage from frost bite, increase the number of outdoor activity hours, and alleviate climatic pressure on urban greenscapes. However, the higher temperatures decrease soil bearing capacity, precursor urban flooding, and enable proliferation of invasive species and diseases (Callaghan et al., 2011).

The UHIs in Arctic cities develop on the background of the already amplified Arctic warming and environmental changes (IPCC, 2013; Lappalainen et al., 2016; Esau et al., 2016). Currently there is no consensus about the magnitude and extent of the Arctic UHIs or the physical mechanisms driving the higher urban temperatures. This is not surprising since, prior to this study, high-resolution in-situ observations have only been reported for the relatively small settlement of Barrow, Alaska (Hinkel and Nelson, 2007) and for southern Finnish (Järvi et al., 2009; Hjort et al., 2016) and some North-European cities (Baklanov et al., 2005; Piringer et al., 2007). Moreover, the larger and more comprehensive surveys of UHIs (e.g. Peng et al.,



2012), which did indicate the increasing magnitude of UHI with latitude (Wienert and Kuttler, 2005), have not extended into the polar regions.

There is a clear and urgent need to improve urban resilience and infrastructure sustainability in cities (Pressman, 2006; Nelson et al., 2001) particularly in the rapidly changing Arctic climate (Streletskiy et al., 2012). This requires closing knowledge gaps on UHIs and equipping city managers with adequate understanding of the challenges of urban planning in cold climates. There is a need to establish the existence, spatial extent, magnitude and physical mechanisms driving wintertime Arctic UHIs. Previous analysis of UHIs have used standards which have been widely criticized (Stewart, 2011), and here we seek to improve on this through synchronized investigation of in-situ and satellite data, exploring the long-term climate statistics for Apatity, and running several model sensitivity experiments with and without an urban scheme. The observational network of the World Meteorological Organization (WMO) is too sparse to capture UHIs in high latitudes. There are no urban WMO stations in Arctic cities. This study reports new results obtained from a unique, dense, in-situ observational network in Apatity – the Urban Heat Island Arctic Research Campaign (UHIARC). The UHIARC network was deployed in Apatity in 2015 and created the opportunity for a comprehensive study of a typical polar UHI combining dense and continuous in-situ observations, satellite remote sensing data, and high-resolution meteorological modelling.

This study reveals a previously under-estimated magnitude of anthropogenic effects on the winter UHI in Apatity. The effects are thought to be similar to those in other polar cities, but they are distinct from the combination and weights of the various causes of UHIs in the more temperate climates of China (Choi et al, 2014; Zhou et al., 2015), Europe (Piringer et al., 2007; Schwarz et al., 2011), and North America (Sailor and Lu, 2004; Zhao et a., 2014).

This study is structured as follows. The next section describes the study area, climate, data sets, and methods. The third section presents the results of statistical data analysis. The fourth section interprets and discusses the results using numerical experiments with the COSMO-CLM model. The final section outlines conclusions of this study. In addition, there is a supplementary material that addresses some technical details of the UHIARC network, data analysis and modelling.

## 2 Data and methods

### 2.1 The study area

Apatity (67.567 °N, 33.393 °E) is a city in the Murmansk oblast (region) of the Russian Federation. It is located in the Kola Peninsula inside the Arctic Circle and so the mid-winter is characterized by a polar-night. The city lies on a low hill amid an undulating landscape between the Imandra lake and the higher Khibiny mountains (Figure 1). The lake was fully frozen during the considered winter periods. The landscape elevations vary from 150 to 200 meters above sea level. Apatity has a cold and humid continental climate (Kottek et al., 2006) with the coldest monthly-mean temperature (January) of $-13.5\,°C$ and a record minimum temperature of $-47\,°C$ observed in 1985. The winter mean temperature is $-8.8\,°C$, which implies persistent snow cover. The annual mean temperature of $-0.4\,°C$ is below the water freezing point, but there are only isolated patches of



permafrost according to the International Permafrost Association (Brown et al., 1997). The warmest summer month is July with an average temperature of $+14.1\,°C$.

The built-up city area comprises $4.5\ km^2$ (excluding the industrial areas). Following the definitions from Stewart and Oke (2012), the major part of the city can be classified as Local Climate Zone 5 (LCZ5) or open midrise buildings while the outskirts of the city could be classified as LCZ7 or lightweight low-rise buildings. The territory of an industrial complex to the north of the city could be classified as LCZ8 (large low-rise buildings) and LCZ10 (heavy industry). The city population was 59 672 inhabitants according to the 2010 census. Apatity has a relatively diversified economy compared with other polar cities, which includes a significant number of jobs in the education and research sector. Nevertheless, as the city name itself suggests, its main industry is organized around production of phosphorus mineral fertilizers.

## 2.2 In situ observational data

Apatity is the only polar Eurasian city that has had a sufficiently dense multi-year urban observational network to comprehensively quantify the Arctic UHI and its driving factors. In-situ meteorological observations in urban and rural areas were collected with a network of *Davis Vantage Pro 2* automatic weather stations (AWS) and with a number of small temperature sensors (*iButton* loggers). The AWSs and loggers were deployed in the city and surroundings (see their description in the Supplementary material S1) during three winters: 2013/14 (from 29/01/2013 to 03/02/2014), 2014/15 (from 29/01/2014 to 04/02/2015) and for a longer period in 2015-2016 (from 10/12/2015 to 28/12/2016). The deployed network of stations, abbreviated here as UHIARC, continues to collect the data. Figure 2 shows the map of the UHIARC and WMO stations in and around the city.

There is only one WMO station in the Apatity region (WMO ID 22213). It is located at the R1 site (at the shore of Imandra lake) 2 km from the city at a lower elevation (132 m above the sea level). The standard meteorological observations at the WMO station were available for all intensive observational periods. The standard WMO temperature data at 2 m height were sampled at 1 min intervals in 2014 and 2015 and at 6-hour intervals in 2015-2016. They were used to run the quality control of the AWS observations. The wind speed at 10 m height and cloud cover measurements were available at 6-hour intervals.

### 2.3 Remote sensing data products

The satellite remote sensing data complement the dense but still highly fragmented in-situ observations. For this study we utilized land surface temperature (LST) data products from the Moderate Resolution Imaging Spectroradiometer (MODIS) sensors aboard the Terra (EOS AM) and Aqua (EOS PM) NASA satellites (Hu and Brunsell, 2015). The MODIS LST data characterize the surface temperature during clear-sky weather conditions. In this sense, the LST data are also incomplete, differ from the in situ air temperature data, and are therefore only complementary to the ground-based observations.

The LST data are in reasonable correspondence to, and rather well correlated with, 2 m air temperatures as was shown in a comparison with observations (see Supplementary material S2). Two MODIS products, MOD11A1 and MYD11A1, supply instantaneous views of the LST at a 1-km resolution, two times a day each (one night-time and one daytime image). Since the


availability of the LST is controlled by objective factors (e.g. cloudiness), we filtered out images with a too large fraction of gaps (more than 20%) within the area. On the remaining images we filled the spatial gaps with the nearest neighbour algorithm and then averaged them over the study period. The documented MODIS LST accuracy is higher than 1°C in the –10°C to +50°C range (Wan et al., 2004).

## 2.4 Sensitivity runs with a limited-area mesoscale meteorological model

We used the limited-area mesoscale meteorological model COSMO-CLM (Rockel et al. 2008) coupled with a bulk urban scheme TERRA-URB (Wouters et al., 2016) to look at the UHI spatial variability and sensitivity to physical drivers. The TERRA-URB scheme accounts for physical features of the urban surface using semi-empirical urban canopy dependences in terms of roughness length, albedo, emissivity, heat capacity, etc. The anthropogenic heat flux (AHF) is calculated according to the Flanner approach (Flanner, 2009). The approach is based on a city-specific annual mean value of the flux, $\overline{Q_H}$, and the prescribed amplitudes of the diurnal and seasonal cycles for the given geographical location. We set $\overline{Q_H}$ to $50\,\mathrm{W\,m^{-2}}$ for the entire urban area and then scaled it for each model grid cell accounting for the cell's urban fraction. It gives $\overline{Q_H} = 90\,\mathrm{W\,m^{-2}}$ for the fully urbanized grid cells during the winter months. Such high values of the AHF are consistent with estimates for other high-altitude cities and with simple estimates for Apatity based on coal consumption data (see Supplementary S3 for details). The COSMO-CLM model with the TERRA_URB scheme has previously been successfully used to simulate UHIs in Belgian cities (Wouters et al., 2016) and in Moscow (Varentsov et al., 2018).

The COSMO-CLM model was used to do nested dynamical downscaling of ERA-Interim reanalysis data (Dee et al., 2011), which is available at 0.75º spatial resolution on 6-hour intervals. Three nested domains were utilized with the spatial grid resolutions of 12 km, 4 km, and 1 km (see Supplementary material S3). The model was run with 50 vertical levels of which 11 levels were in the lowest 1 km of the atmosphere. The model top was set at the 20 hPa level. The model simulations were conducted for the winter season of 2015/16. They included three separate numerical experiments: the URB_AHF run with the TERRA_URB scheme switched on and the AHF defined as described above; URB_noAHF with TERRA_URB switched on and zero AHF; noURB run with TERRA_URB switched off and zero AHF.

## 3 Results

### 3.1 Dynamics and controlling factors of the Arctic UHI

UHIARC observations show the strong and persistant urban temperature anomaly for all three winter periods (Figure 3). The urban stations are always warmer than the rural ones, although the magnitude of this positive anomaly strongly varies depending on the weather conditions. Analysis in this study is based on a temperature difference $\Delta T_j^i = T_i - T_j$. The UHI intensity is defined with respect to the R1 site (the WMO station) as $\Delta T_{R1}^{U1} = T_{U1} - T_{R1}$. This temperature difference represents the deviation of the measured temperature in the city center (the U1 site) from the measured temperature at the WMO station





(the R1 site). The WMO station is used as a baseline source of weather- and climate-related information in the studied area. However, R1 and U1 sites are found at different elevations (132 and 180 m above the sea correspondingly) are situated dinfferently with respect to the local orograhy features. Besides, the WMO station is situated close to the Imandra Lake, but our analysis consider only winter conditions with a frozen lake surface, so the influence of the water area can be excluded.

Below, we will examine the effects induced by the 48 m elevation difference and orograhy effects as well as by the anthropogenic UHI drivers.

Co-variability between the UHI intensity and the controlling meteorological variables such as wind speed, $u$, and low and total cloud fraction, $n_l$ and $n_t$ respectively, is presented using a weather factor, $W_f$. This index was adopted from Oke (1988) and made it applicable to a set of observed parameters available in the WMO station data:

$$W_f = min\left(1, u^{-\frac{1}{2}}\right) \cdot (1 - 0.8n_l^2 - 0.4(n_l - n_t)^2) \tag{1}$$

The high values of $W_f$ correspond to calm (anticyclonic) weather and clear sky conditions. Specific climate conditions of high latitudes, namely, low winter temperatures, deep snow cover and almost absence of direct solar radiation, make many proposed UHI scalings(e.g. Theeuwes et al., 2016) inapplicable for this study. During two short observational periods, the sun was above the horizon less than 5 hours per day with the maximum solar angle being less than 5°, and so the direct shortwave radiation flux was close to zero. The long-term observations cover even the polar-night period when there is an absence of direct solar

radiation. The small or absent shortwave radiation flux in all three observational periods results in a negligible diurnal temperature cycle as the temperature varies irregularly following the synoptic variability.

There were large differences in the synoptic conditions during the considered three winters. There was a strong anticyclone for most of the first observational period from 26th Jan to 1st Feb 2014 (Konstantinov et al. 2015). The persistent clear-sky weather in the anticyclone led to intense radiative cooling and the surface air temperature dropped to -22 °C in the city and to

–27 °C and –30 °C in the nearby suburbs. A strong atmospheric temperature inversion developed with the temperature increasing by 20K across the lowermost 1 km of the atmosphere. During this period, the average of the observed UHI intensity exceeded 5 K with a maximum of 10 K observed on 31/01/2014. In contrast, there were usually cyclonic conditions with notable wind gusts and multi-level cloud cover during the second observational period at 2015. Nevertheless, a brief period of calm and clear conditions (from 31/01/2015 to 02/02/2015) led to strong radiative cooling and a rapid increase in the intensity of

the UHI.

The third winter with the longest observations (from 10/12/2015 to 28/01/2016) was characterized by several periods of anticyclonic conditions of which the four longest, with $\Delta T_{R1}^{U1}$ reaching up to 11 K, were selected for more detailed case studies (see Supplementary material S4). The mean value of $\Delta T_{R1}^{U1}$ over the winter period was 1.9K with 95%-quantile of 7.8K. The values higher than 5K were observed 20% of time. The largest UHI events occur during periods with $W_f > 0.7$. The maximum

UHI intensity depends on the persistency of the anticyclonic conditions with the large $W_f$.

Using the historical WMO records for the R1 site we were able to quantify the overall importance of the large winter UHI in the urban climatology. Figure 4 illustrates that the high frequency and long-term persistency of the cold spells in the polar





areas – the heavy tails in the temperature distribution that are a distinct feature of the polar continental climates (Timlin and Walsh, 2007) – make the conditions with an intense UHI rather common and therefore regularly impacting human life and urban management. A strong linear regression is found between mean (maximum) $\Delta T_{R1}^{U1}$ values and the daily mean (minimum) temperatures, with $R^2 = 0.71$ (0.69) and $p < 0.01$. The largest contribution to the regression, and to the observed UHI, is

provided by the observations at the lowest temperatures, typically below –10 °C. The probability distribution of the observed temperatures shows that such low temperatures occur rather frequently in the winter months.

The urban temperature anomalies calculated in relation to other rural sites were also positive, but less pronounced. We note that $\Delta T_{R2}^{U1}$ and $\Delta T_{R5}^{U1}$ are about half as strong as $\Delta T_{R1}^{U1}$ (see Supplementary materials S5 and S6). The stations U1, R2 and R5 are found at elevations 180, 155 and 170 m correspondingly. They are situated in similar landscape with respect to the local

orographical features. Thus, those observations suggest that the other factors, primarily the anthropogenic heat flux (AHF), should create about one half of observed $\Delta T_{R1}^{U1}$ values. The other half of the difference is likely induced by the elevation difference and local orograhy features.

### 3.2 Spatial structure of the observed temperature anomalies

The dense UHIARC network revealed that the positive temperature anomalies are indeed located within the built-up areas

(Figure 5). Moreover, the larger positive $\Delta T_{R1}^{i}$ are collocated with the densely built, central parts of the city. The large number of rural sensors in 2015 show the generally lower temperatures found outside the urban environment, and particularly in the lower-altitude locations.

To gain a broader and more consistent, albeit less detailed, view on the spatial temperature field in the area, we performed analysis of MODIS land surface temperature (LST) data. This analysis showed that the positive temperature anomalies are not

only collocated with the urbanized hill where the city is located, but also over the other hills in the area including the higher Khibiny mountains (Figure 6). The LST contrasts are even higher during anticyclonic periods with high weather factor (see Figure S6). The observed spatial distribution of LST confirms that the regional-scale temperature inversion is a significant factor in determining the LST variability over an undulating surface in polar areas which results in a clear relationship between the LST and the elevation of the surface(Figure S7). Despite this clear tendency for the higher locations to be warmer than

lowlands, urbanized and industrial areas are warmer then the most of non-urban areas located at the similar elevation. Only the southern slopes of the Khibiny mountains, which are covered with darker and denser coniferous forest, exhibit warmer LST than urban or industrial areas located at similar elevation.



## 4 Discussion

### 4.1 Driving factors of the strong UHI in Apatity

The analysis of both in-situ and satellite temperature data have shown a strong and persistent winter temperature anomaly that is collocated with the urbanized area of Apatity. The winter values of $\Delta T_{R1}^{U1}$ are comparable with the UHI intensity observed

in Moscow (Lokoschenko et al., 2014; Varentsov et al., 2018), London (Wilby, 2003), New-York (Gedzelman et al., 2003), Beijing, Seoul (Choi et al., 2014) and other world megacities. This warm anomaly is particularly intense in the coldest weather conditions, but it becomes much weaker under stronger winds. As the polar climate frequently experiences anticyclonic weather, significant and persistent warm temperature anomalies are ubiquitous during wintertime in Arctic cities (Figure 4). Although the UHI has been reported for other polar cities as well, their physical drivers remained unclear. In contrast to mid-

and low-latitude cities, the polar night conditions exclude the urban trapping of solar radiation and lower evapotranspiration rates (Ruy and Baik, 2012) from considerations as UHI driving factors. In addition, the solar heating deficit causes a cooling of deforested surfaces in high latitudes (Lee et al., 2011), which could be also expected for mostly deforested urban areas. Nevertheless, our urban data firmly establish the emergence of pronounced warm anomalies under calm, clear-sky weather conditions.

Such a pattern is compatible with two different, but not necessarily alternative, physical drivers proposed in the literature. Firstly, a UHI could be caused by direct anthropogenic factors, including the heat released from buildings and industrial processes, which would explain why the largest temperature anomalies were observed in the most built-up areas of the city. Alternatively, the warm anomaly could be caused by elevated orography, such as the hill on which Apatity is sited, under conditions of near-surface atmospheric temperature inversions, which is a common meteorological phenomenon in the high

latitudes (Wetzel and Brummer, 2011), favourable for cold air drainage to lowlands (Daly et al., 2014). Indeed, the remote sensing data show warmer land patches on the surrounding territory with the temperature anomalies being strongly correlated with surface elevation (Figure 6, S6, S7).

To clarify the relative contribution of the different physical drivers, we ran several sensitivity experiments with the high-resolution meteorological model COSMO-CLM (see Data and Methods section and the Supplementary material S3). The finest

resolution (1 km) model run reproduces the local temperature and wind variability in the considered domain reasonably well (Figure S8). The model performance deteriorates under calm weather conditions with a high weather factor, $W_f$. This well known, but still poorly understood, effect is traced to the breakdown of the turbulence closure schemes under stable stratification (Fay and Neunhäuserer, 2006; Baklanov et al., 2011; Atlaskin and Vihma, 2012; Davy and Esau, 2014). In the model, the temperature tends to cool significantly slower than in observations under the conditions when the UHI is expected

to be the largest. Despite these model inaccuracies, the COSMO-CLM model with the TERRA_URB scheme and a prescribed AHF (URB_AHF run) was able to simulate the UHI (Figure 7).

The UHI intensity in the model is defined as the difference between the temperatures of the model grid cells which include the U1 and R1 stations. The UHI intensity was 2-5 K or about 50 % of the observed $\Delta T_{R1}^{U1}$ for the majority of the anticyclonic





events with $W_f$ >0.7. The mean $\Delta T$ over the simulated period of the winter 2015-2016 was 0.8 K with the 95 % quantile of 3.2 K. The observed values were 1.9 K and 7.8 K correspondingly. The correlation coefficient between the simulated and observed $\Delta T$ is 0.6 (p<0.05). This overall underestimation of the UHI intensity (Figure 7), cooling rates and the weather factor $W_f$ (Figure S8) could be partially due to the cloud cover overestimation, which is typical for the COSMO model (Jaeger et al.,

2008; Pfeifroth et al., 2012). Moreover, the model does not take into account that anthropogenic heating intensifies during the coldest conditions, as the coal consumption increases (see the data in Figure 10). Finally, the model resolution (1 km) seems to be insufficient to resolve the local air circulation between the hills, which occupy 1-2 model grid cells.

We compared these results to those runs without the TERRA_URB scheme (noURB run), i.e. when the model was not provided the information about the anthropogenic heat fluxes and specific urban land use – land cover types, and runs with the

TERRA_URB scheme, but without the AHF (URB_noAHF run). These additional runs were unable to reproduce the UHI. The mean $\Delta T_{R1}^{U1}$ in the runs noURB and URB_noAHF were –0.25 K and 0.06 K respectively with the 95 % quantile of 0.85 K and 1.5 K. The correlation coefficients were 0.36 in the URB_noAHF and nearly 0.0 in the noURB runs.

To investigate the geographical pattern of the temperature sensitivity we chose 6 shorter case studies of very intense observed UHIs, which persisted for over 150 hours in total. The model output and the observations agreed rather well for those 6 cases

(Figure 7 and Supplementary Figure S8). Mean values of $\Delta T_{R1}^{U1}$ and $\Delta T_{R5}^{U1}$ over all 6 cases are 5.3 K and 2.9 K correspondingly. The UHI over Apatity is clearly distinguishable in the URB_AHF simulations where four patches of lower temperature can be distinguished (Figure 8). Whereas in the noURB simulations the cold temperature patches are connected and fully occupy the lowland area including the Apatity hill, while the other hills and mountains remain warm. These results indicate that orography-induced circulations are likely under-resolved in the simulations.

The differences between the runs with and without the urban scheme (i.e., URB_AHF – noURB and URB_noAHF – noURB) demonstrate that the temperature anomalies due to both the AHF and the land use – land cover modification are collocated with the urbanized areas (Figure 9 and S9). Such properties of the UHI are consistent with the widely accepted explanation that the additional urban heating is trapped within the city in the shallow and stably-stratified atmospheric boundary layer (Baklanov et al., 2005, 2008; Piringer et al., 2007; Davy and Esau, 2016). This exercise showed that the AHF is the dominant

factor driving the winter UHI. Indeed, the URB_AHF run was 2.5 K warmer in the central grid cell with the U3 site than the noURB run. The difference between the URB_noAHF and noURB runs was only 0.6 K.

Coherent evidence from the modelling results, in situ and remote sensing data indicate that the anthropogenic contribution to the observed urban temperature anomaly in Apatity ($\Delta T_{R1}^{U1}$) is about 50% (and likely more in the very cold days). The rest of the anomaly is caused by the local orography variations. The sensitivity model runs confirmed that the dominant fraction of

the anthropogenic contribution comes directly from the AHF. This contribution is larger in the central urbanized areas, but it could also be identified in the surrounding non-urban areas, which are 2-3 times the size of the built-up area. Similar and even more pronounced results have been reported in large-scale studies of Chinese cities (Choi et al., 2014; Zhou et al., 2015).



## 4.2 Impacts of a strong UHI in Arctic cities

The model and observations both show that there is a considerably milder winter micro-climate inside Apatity and the largest difference with the surroundings is found during the coldest periods. This intense UHI may cause several environmental and socio-economic effects, not all of them are necessarily detrimental. Among the most impactful effects are the higher soil
temperatures and deeper active soil layers. Although there is no permafrost around Apatity, a number of medium-sized or bigger cities (e.g. Nadym, Novy Urengoy and Norilsk) with similar urban planning exhibited strong signed of degrading permafrost (Streletskiy et al., 2012). The northern ecosystems and soil micro-biota are also sensitive to this warmer micro-climate. The city becomes a corridor for invasive southern species that were partly introduced on purpose with the intention of creating urban green space (Byalt and Byalt, 2011) with blossoming trees and bushes.

The UHI can also trigger a direct economic effect. Warmer temperatures increase the number of outdoor working hours and decrease the risk of frostbite for city dwellers. Such conditions are also welcome for outdoor recreational activities. Considering the urban metabolism, Figure 10 shows that the daily coal consumption for anthropogenic heating in the city strongly depends on the air temperature. If the UHI effect was fully accounted for in the production and distribution of heat, average coal consumption in the city could be reduced by about 1.5%. Following this linear relationship of 33 tons of coal consumption per
$1^{\circ}C$ of temperature change, coal consumption could be reduced by 12% in the coldest days when consumption is in its highest. A more flexible heating schedule and infrastructural improvements would not only be more economical, but also a climate change mitigation action of the municipality.

Studies of urban socio-environmental phenomena, pollution from urban sources and stability of frozen soils are of high importance in the strategical implementation plan of the Pan-Eurasian Experiment (PEEX) as stated in Lappalainen et al.
(2016) and Melnikov et al. (2018). The clear-sky events are not only causing the strongest deviations of the local urban climate, but also affect particle nucleation (Dada et al., 2017). Processes of turbulent mixing of atmospheric pollution in cold winter conditions remain to be in focus of studies for the international research community (Arnold et al., 2016; Kim et al., 2017). Moreover, the warmer climate conditions are suspected to fundamentally change vegetation responses and carbon storage processes in the Arctic soils (Arneth et al., 2016). Those processes could be already observed and hence observationally studied
on territories covered by urban heat islands.

## 5 Conclusion

The model sensitivity study, unique new in-situ observations, and remote sensing data analysis presented here comprise the first comprehensive analysis and attribution of the Arctic UHI effect during the polar winter. Understanding of the driving factors behind UHIs is urgently needed to inform urban planning and management, and to optimize heating and energy
consumption. Here we have shown that the largest contributing factor to the UHI in the typical medium-size polar city of Apatity was from direct anthropogenic heating effects and land-use modifications, which could be additionally amplified by the local orography and presence of atmospheric inversions.



Wintertime Arctic UHIs can be much stronger than those found in mid-latitudes. The wintertime average temperature difference between the Apatity city center and the local WMO weather station is 1.9 K with extremes up to 11 K and excess of 5 K occurring in one in five days in winter. At least 50% of this warm anomaly is caused by the UHI effect, driven mostly by direct anthropogenic heating. This means that the anthropogenic urban heat fluxes warm the city by about 1 K on average in

wintertime. The AHF during extreme cold days may warm the city center by up to 6 K. The UHI is more intense in the cold and calm days that were frequently observed during polar anticyclonic intrusions. The warm anomalies are larger in the central parts of the city but the UHI also has an impact on the non-urban surroundings. The high latitude regions and the Arctic experience intense warming. The urban temperature anomalies add an additional contribution on top of this regional warming. This strong Arctic UHI makes the urban microclimate significantly milder, reduces soil freezing, and changes the soil

hydrology. This can put buildings at risk, which is a widespread problem for more than 50 similar and larger cities in the Arctic. Many of them are located in more severe climates and have been constructed on artificial sandy ground or permafrost that is highly sensitive to warming (Nelson et al., 2001; Streletskiy et al., 2012). The physical impact of the UHI also brings economic and ecosystem impacts. The mild microclimate inside Arctic cities can act as a corridor for invasive species, paving the way for broader regional ecosystem changes in a warming climate. The negative impacts of this large UHI clearly show

that the construction, maintenance, and energy-efficiency standards need to be significantly improved if we are to achieve sustainable city planning in this sensitive environment. This is a strong argument in support of energy efficiency efforts and against high-density urban development in polar settlements. However, not all UHI impacts are negative, and our study also illustrates a pathway to take specific climate change mitigation actions. The consumption of coal for urban heating could be substantially reduced if the UHI was accounted for in local urban management.

**Data availability:** The UHIARC datasets and modelling results generated during this study are available from the corresponding author on reasonable request.

**Author contributions:** I.E. and R.D. wrote the paper and guided the data analysis. A.B., P.K., M.V. and I.E. worked on the idea and the hypothesis of the study. P.K. and M.V. collected and processed the data, prepared illustrations and tables. M.V. ran the model and analysed model results. V.M. and M.V. processed the satellite data. A.B., I.E. and V.M. provided the socio-
environmental context to and described the impact of the study. R.D., V.M., P.K and M.V. provided the physical climatic context for the study. P.K., V.M., I.E., and A.B. discussed and interpreted the results.

**Conflicts of interest:** The authors declare no conflict of interest.

**Acknowledgements:** This study was supported by the Belmont Forum and the Norwegian Research Council grant HIARC: Anthropogenic Heat Islands in the Arctic: Windows to the Future of the Regional Climates, Ecosystems, and Societies (no.
247268) and by the Russian Foundation for Basic Research (RFBR) projects no. 15-55-77004, 17-05-01221, 18-05-60126 and 18-05-00715 and by the project TRAKT-2018 (Transferable Knowledge and Technologies for High-Resolution Environmental Impact Assessment and Management). The first stage of experimental campaign in Apatity in 2014 was supported by Russian Geographic Sosciety project no. 27/2013-H3. The numerical simulations with COSMO-CLM model are carried out using the equipment of the shared research facilities of HPC computing resources at Lomonosov Moscow State University.




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



**(a)**                                          **(b)**

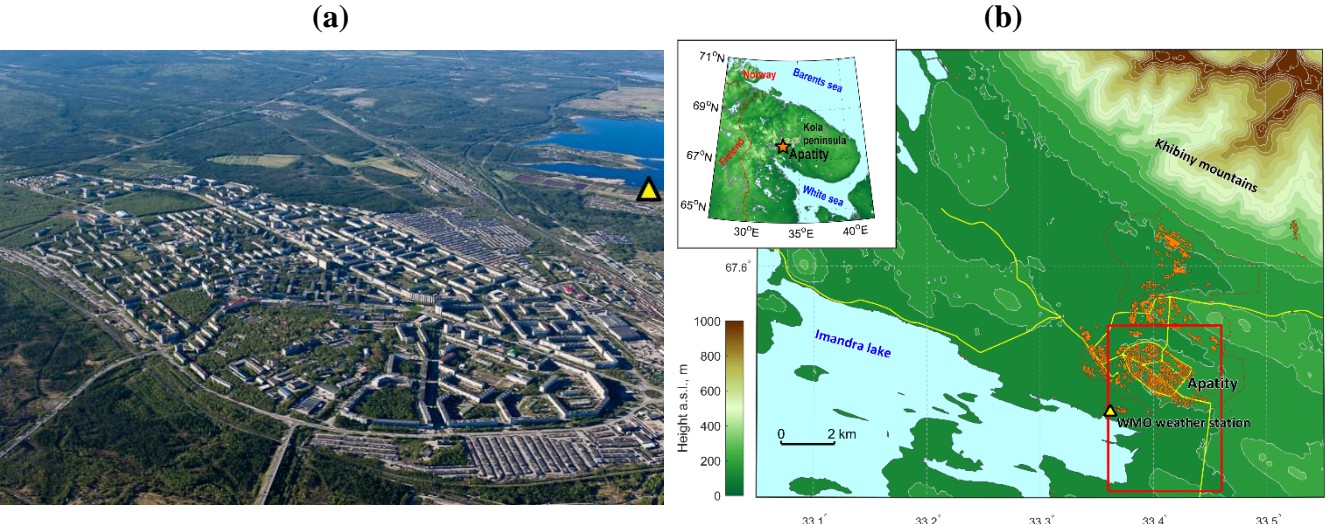

**Figure 1.** (a) An aerial view of Apatity in summer taken from the northern side of the city (photo credit: B. Vakhmistov); (b) relief map of Apatity. Colours show the elevation above sea level. Buildings are indicated in orange, major roads by yellow lines and the red dotted line shows the city administrative border. The red rectangle corresponds to the area shown in the aerial view (a). The yellow triangle in (a) and (b) marks the location of the WMO weather station R1.



**(a)**

**(b)**

**(c)**

**Figure 2.** Maps of the observation sites from the (a) 2014, (b) 2015, and (c) 2015/16 field campaigns. The details of the location and a description of individual sensors are given in Table S1. Elevation is shown by green isohypses, which are indicated at each 25 meters, with the isohypses of 150 and 200 meters shown by bold lines.





**Figure 3.** Observations of the surface air temperature in the city centre (the red line for the station U1) and the rural sites outside the city (the black, blue and green lines) in the winters (a) 2014, (b) 2015 and (c) the longer observations in the winter 2015-2016. The UHI magnitude (area shaded by orange) is defined as $\Delta T_{R1}^{U1}$. For the 2014 and 2015 data, 30-minute running means are shown, whereas 3-hour running means are shown for the longer experiment over 2015-2016. The six case study periods indicated in black were selected for further study (see Figures 4 and S4). The weather factor $W_f$ (the blue line on the lower panel) was calculated from observations at R1 using 6-hourly data. The solar angle is indicated by the orange line with shading indicating the periods with positive values. The light blue shading indicates the anti-cyclonic weather periods with $W_f > 0.7$. The geographical locations of the stations are shown in Figure 2.





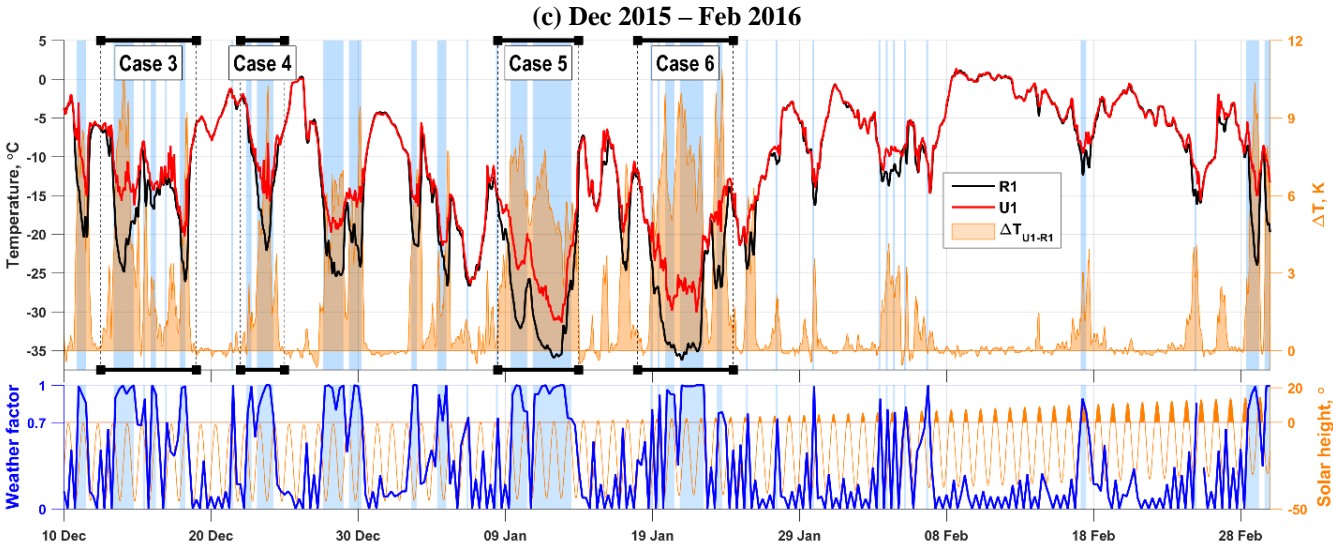




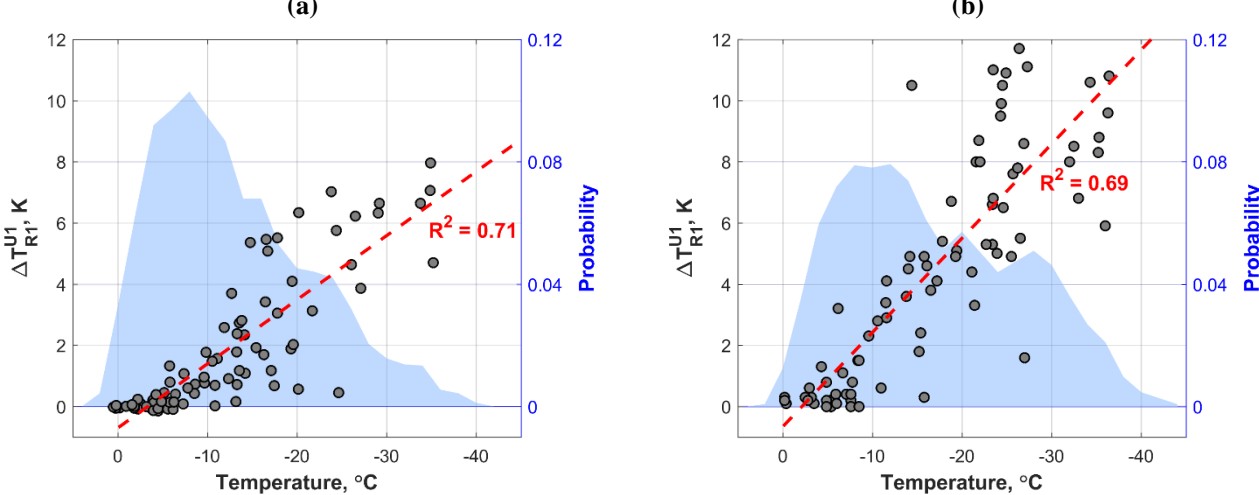

**Figure 4.** The observed daily mean (a) and daily maximum (b) temperature difference $\Delta T_{R1}^{U1}$ (circles) as a function of (a) the daily mean and (b) the daily minimum temperature at the R1 station during the winter of 2015-2016. The red line shows the linear regression. The shaded area shows the probability distribution of the (a) daily mean and (b) daily minimum temperature at the R1 station for the winter months (DJF) of 1986-2016.





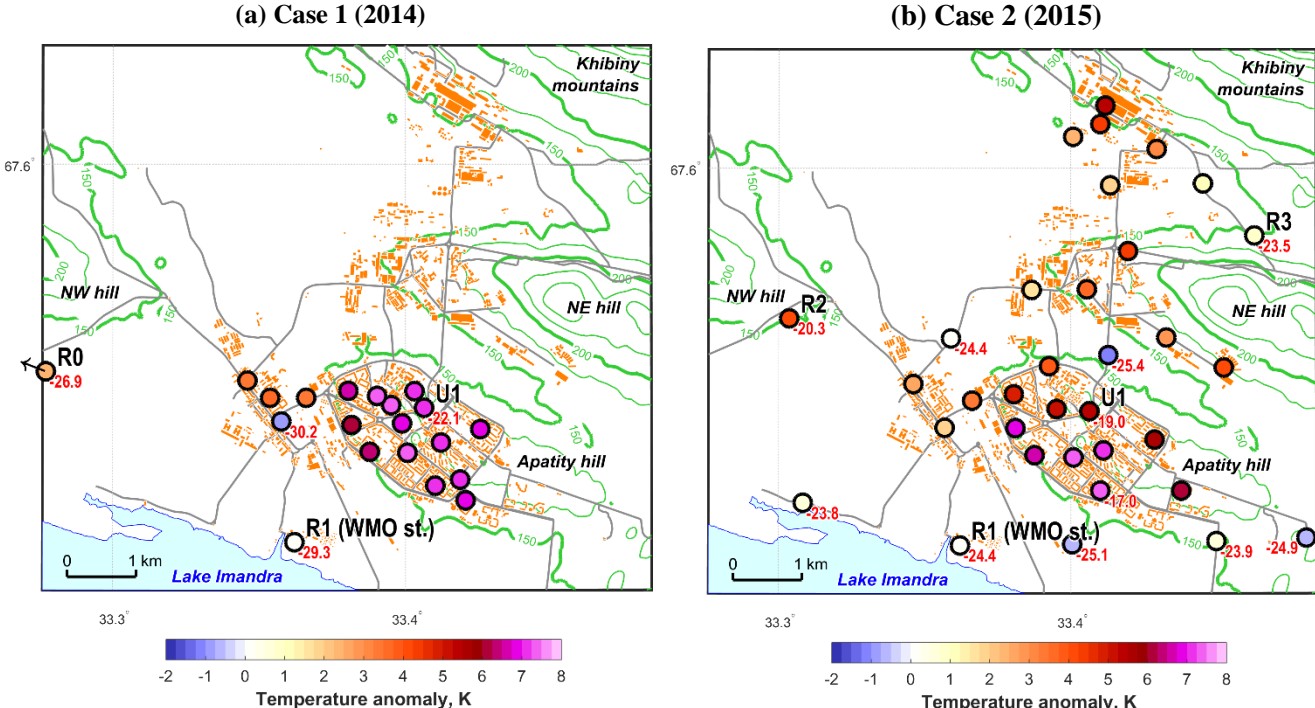

**Figure 5.** Spatial patterns of the air temperature anomalies at each station with respect to the temperature at the R1 station. The anomalies were averaged over 3-hour time intervals which includes the maximum temperature anomaly during the observation periods of (a) 2014 (Case 1 from Figure 3) and (b) 2015 (Case 2). The colour scales show the magnitude of the temperature anomaly. The red numbers indicate the absolute temperatures in °C. The temperatures were measured at a height of 2 m above the ground, i.e. they correspond to significantly different elevations above the sea level. The surface elevation is shown by the green isohypses (drawn every 25 meters), and the isohypses of 150 and 200 meters are drawn in bold.





**(a) MODIS AQUA (DJF 2011-2016)**  **(b) MODIS TERRA (DJF 2011-2016)**

**(c) MODIS AQUA (DJF 2015-2016)**  **(d) MODIS TERRA (DJF 2015-2016)**

**Figure 6.** The mean winter (DJF – from December to February) land surface temperature (LST) anomaly obtained from satellite remote sensing with the MODIS instruments onboard the TERRA and AQUA satellite platforms for (a,b) the longer period of 2011-2016 and for (c,d) the period of 2015-2016, which is synchronized with the long-term UHIARC in-situ measurements in Apatity. The LST anomaly is calculated as the difference in temperature from the temperature of the pixel collocated with the R1 station. The red lines encompass the pixels classified as the urban area; the yellow lines – the industrial area; the cyan lines – the Khibiny mountain area.



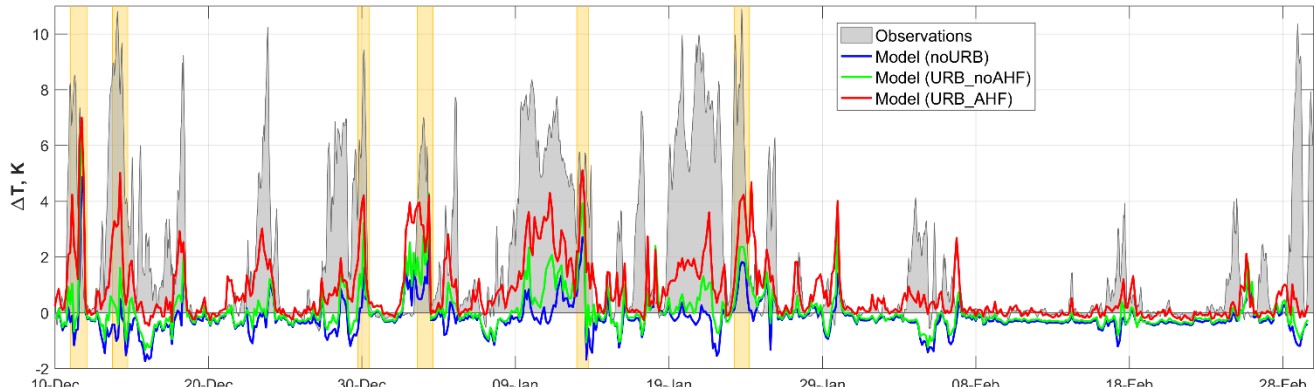

**Figure 7.** Intercomparison of the observed (gray shading) and modelled (the red, green and blue lines for URB_AHF, URB_noAHF and noURB runs correspondingly) UHI intensity. The UHI intensity is taken as $\Delta T_{R1}^{U1}$ for the observations and the corresponding $\Delta T$ values for the grid cells collocated with the U1 and R1 sites in the model. The model output was sampled at 3-hour intervals. The orange shading identifies the cold periods with strong UHI which were selected for sensitivity studies.





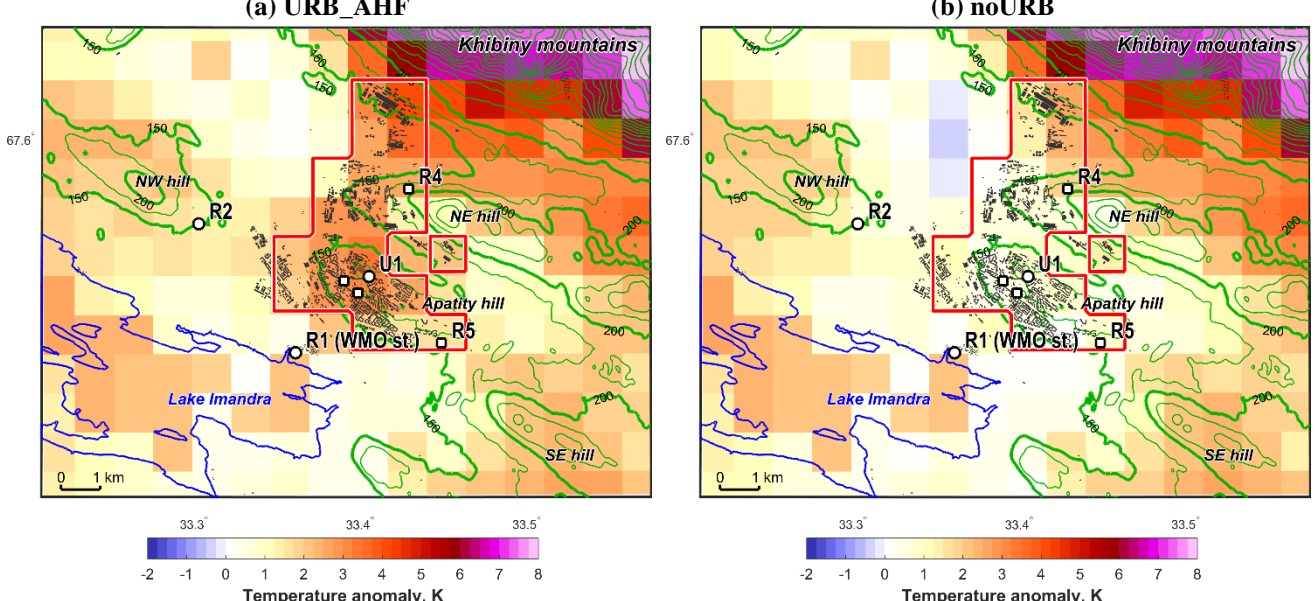

**Figure 8.** The maps of the simulated surface air temperature anomalies, averaged over all six selected cold periods as indicated in Figure 7. The COSMO_CLM simulations with the TERRA-URB urban scheme and prescribed anthropogenic heat flux (URB_AHF) is shown in the left panel (a), and those without the urban scheme (noURB) in the right panel (b). The anomalies were defined as the deviation from the model grid-cell corresponding to the R1 site. The urban land use – land cover was introduced in the grid cells encompassed by the red borderline.





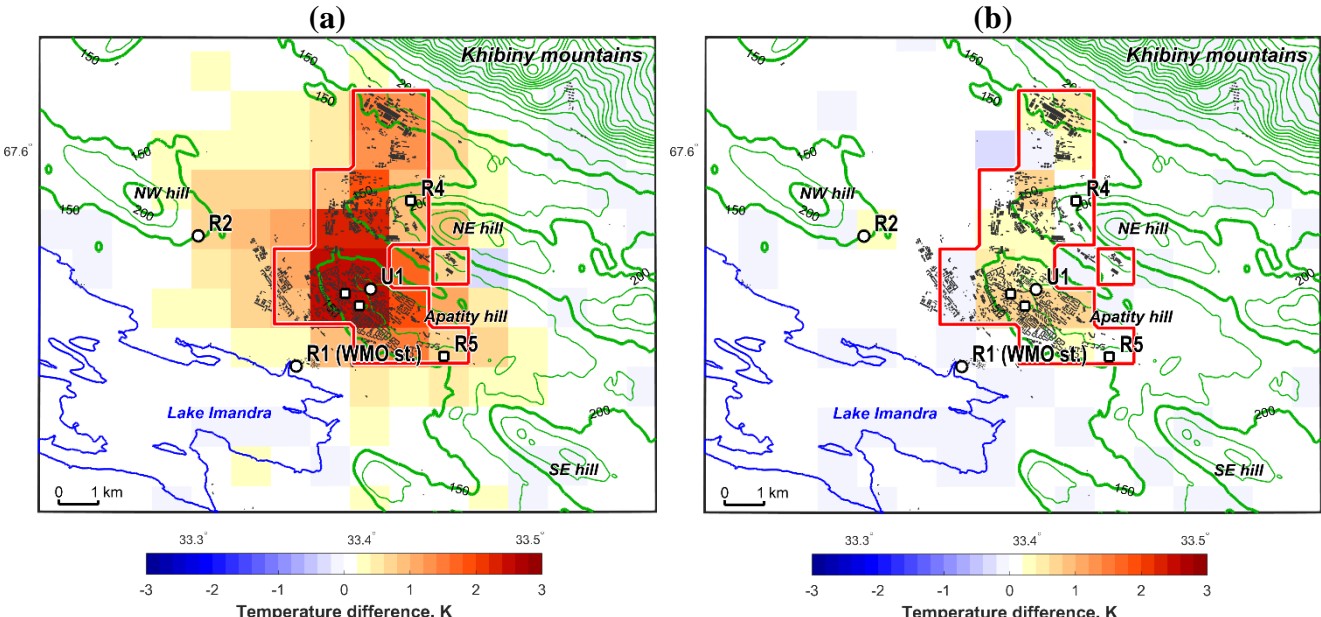

**Figure 9.** Maps of the simulated surface air temperature difference between the COSMO-CLM runs: URB_AHF – noURB (a), URB_noAHF – noURB (b). The differences are averaged over the six selected periods identified in Figure 7. Similar differences averaged over the whole winter are shown in the Figure S9. The urban land use – land cover was introduced in the grid cells encompassed by the red lines.



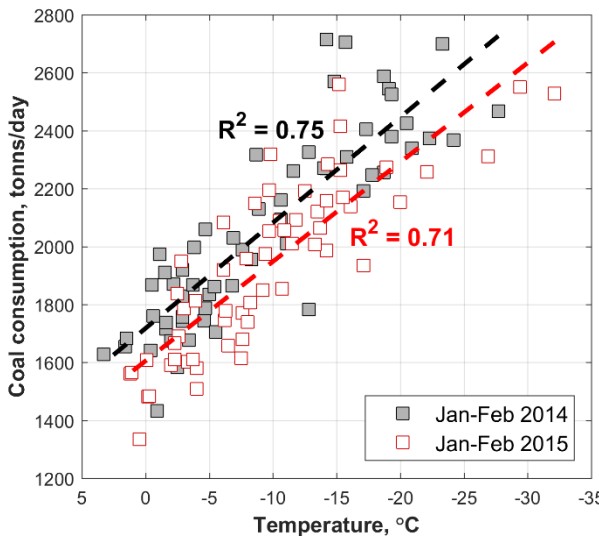

**Figure 10.** Daily coal consumption as a function of the mean air temperature for the winters of 2014 and 2015. Data were provided by the administration of Apatity combined heat and power plant.

