# Peer review of "Anthropogenic and natural drivers of a strong winter urban heat island in a typical Arctic city"

_Atmospheric Chemistry and Physics, 2018_

## Referee Comment (RC1) · Anonymous Referee #2 · 1 Oct 2018

The paper presents results of an unique observation dataset, obtained in the Russian polar area, specifically in the Apatity city and its surroundings. Observed data together with satellite measurements and high-resolution model results give an interesting view in the urban heat island in polar region. I have following comments to the text:

Specific comments:

1) Page 2, line 15: The winter UHI in mid-latitude cities is not so environmental problem, but it can be significant and also dependent on the anthropogenic heat release. E.g., Bohnenstengel et al. (2012) conclude impact of AH to 1.5 K in December in London. Also model studies (Trusilova et al., 2016; Huszar et al., 2014) found the winter UHI in central European cities.

[Figure]

2) P. 5, l. 23: What does it mean TERRA_URB switched off – removing of urban fraction from model grid-boxes (annihilation approach, e.g. Baklanov, 2016) or only not using of TERRA_URB parametrization? The second option admits that still some physical properties of the surface are altered for urban grid cells (by default in the model).

3) P. 7, l. 10: The conclusion about AHF is too fast. Despite a very low solar radiation, the UHI is not created only by AHF. E.g., a reduced long-wave radiation in the urban environment (due to reduced sky-view factor) can also contribute to the UHI formation in calm anticyclonic situations. Similarly in discussion (p. 8, l. 10-11), the driver could be mentioned.

4) P. 8, l. 5: It is inaccurately to attribute the temperature difference between U1 and R1 as the "UHI intensity" and moreover, compare it with values for listed megacities. The U1-R1 difference is created not only by anthropogenic factors. Only about 50 % (1 K) is caused by UHI effects, as correctly written in the conclusion.

5) P. 9, l. 13: What is the base of the selection of six cold periods? The observed U1-R1 differences are sometimes higher and more persistent in other periods (Fig. 7), e.g. after 9th or 19th January. Need to clarify.

6) P. 11, l. 5: There is no clear evidence for the conclusion "The AHF during extreme cold days may warm the city center by up to 6 K". The 50% contribution of all anthropogenic impact is in the time average (as you wrote in p. 9., l. 28).

7) Abstract: The values of 1.9 K and 11 K are misleading, because there is no information about different altitudes, which has a significant impact (of the same magnitude as the anthropogenic impact).

8) Abstract: The sentence "... direct anthropogenic heating contributes at least 50% to the observed UHI intensity, and the rest is created by natural microclimatic variability..." is wrong, because the UHI intensity can not be created (in principle) by "natural micro-climatic variability...". The true sentence is "At least 50% of this warm anomaly

[Figure]

ACPD

---

## Author Comment (AC1) · 24 Oct 2018

We thank the Anonymous Referee for careful reading of our discussion paper. We find a number of his/her suggestions to be very useful for our study and will revise the manuscript accordingly after the end of public discussion. Detailed responses to each of the comments is presented below and duplicated in the attached PDF file.

Comment: The paper presents results of an unique observation dataset, obtained in the Russian polar area, specifically in the Apatity city and its surroundings. Observed data together with satellite measurements and high-resolution model results give an interesting view in the urban heat island in polar region.

Response: we would like to thank the Referee for so high appreciation of our work,

that inspires us for further improvements of our study and further developments of this research direction.

Specific comments:

Comment: Page 2, line 15: The winter UHI in mid-latitude cities is not so environmental problem, but it can be significant and also dependent on the anthropogenic heat release. E.g., Bohnenstengel et al. (2012) conclude impact of AH to 1.5 K in December in London. Also model studies (Trusilova et al., 2016; Huszar et al., 2014) found the winter UHI in central European cities.

Response: We agree with the Referee that the UHI in the mid-latitude cities could be significant and intensive also in winter months. Although the wintertime UHI has been recognized in the studies referred to by the Referee and in our manuscript, there were relatively little attention to it so far. Our study does not object the existence of the wintertime UHI in the mid-latitude cities. But it maintains that the winter UHI is not an important environmental problem in contrast to the summer UHI there. We specifically mention that the mean winter UHI is less intensive than that mean summer UHI as the referred studies have demonstrated.

Comment: P. 5, l. 23: What does it mean TERRA-URB switched off – removing of urban fraction from model grid-boxes (annihilation approach, e.g. Baklanov, 2016) or only not using of TERRA-URB parametrization? The second option admits that still some physical properties of the surface are altered for urban grid cells (by default in the model).

Response: The noURB run follows the annihilation approach. The TERRA-URB scheme was switch off AND the urban fraction was removed from model grid cells. The urban fraction in noURB run was set to zero. The land-cover parameters for the urban grid cells were set equal to the nearest non-urban grid cells. We will clarify the issue and will refer to the annihilation approach (Baklanov et al., 2016) in the text.

Comment: P. 7, l. 10: The conclusion about AHF is too fast. Despite a very low solar radiation, the UHI is not created only by AHF. E.g., a reduced long-wave radiation in the urban environment (due to reduced sky-view factor) can also contribute to the UHI formation in calm anticyclonic situations. Similarly in discussion (p. 8, l. 10-11), the driver could be mentioned.

Response: We agree with the Referee that the AHF scale and impact need further investigation. Concluding the analysis of observations, we formulated a hypothesis about the impact of the AHF on the UHI. We study this hypothesis later in the manuscript, presenting the results of the sensitivity experiments with COSMO-CLM model. In the text, we emphasize that there are other plausible drivers of the UHI in this city. Those drivers, such as long-wave radiation and sky-view factors, have not been addressed in this study. Nevertheless, the model experiments suggest that there is not a big heat budget imbalance to be attributed to other factors then the AHF.

Comment: P. 8, l. 5: It is inaccurately to attribute the temperature difference between U1 and R1 as the "UHI intensity" and moreover, compare it with values for listed megacities. The U1-R1 difference is created not only by anthropogenic factors. Only about 50

Response: We would like to highlight that we specifically avoid calling the temperature difference between the U1 and R1 sites as the UHI intensity in our study. However, such pair of stations characterizes the difference between the city and the nearest WMO station. Without specific knowledge about local microclimates, such difference is often associated with UHI intensity. Strictly speaking, we do not compare our data with the numbers given for the biggest megacities. We used those numbers to put the study into a recognized context. We show to important facts in the perceived comparison: 1) there are very few (typically just one) relevant meteorological stations to get climate information for a medium city, such as Apatity, and this station is not representative for urban environment; 2) there is a surprisingly strong urban-rural temperature difference in Apatity, which is a motivation for further investigation of the driving factors of such

difference.

Comment: P. 11, l. 5: There is no clear evidence for the conclusion "The AHF during extreme cold days may warm the city center by up to 6 K". The 50

Response: We do not agree with the Referee. Our observations and modelling results, presented in this study, show that the given estimate of 50

Comment: 7) Abstract: The values of 1.9 K and 11 K are misleading, because there is no information about different altitudes, which has a significant impact (of the same magnitude as the anthropogenic impact).

Response: The given numbers shows the apparent observed temperature difference. Nowhere in the manuscript this difference is referred to as the UHI intensity. Moreover, we investigated what part of this observed difference could be attributed to the UHI intensity within the limitations imposed by the observation network, methodology, period of study and the limitations of the utilized modeling experiments. This context is clarified further down in the Abstract. At the same time, we do agree that those numbers are hooking a potential reader and call for more extensive research. Comment: Abstract: The sentence "... direct anthropogenic heating contributes at least 50

Response: We agree with Referee about this point. We will follow his/her advice and correct this sentence accordingly.

Technical corrections:

Comment: P. 3, l. 15: The sentence and all paragraph (comparison with other studies) belongs rather to discussion. In this part of introduction, aims of study should be specified.

Response: The aim of the study specified above, see p.3 l. 11. We will separate this part of introduction to a separate paragraph. And the paragraph, indicated by Referee, refers to the main findings of the study in the context of state-of-the-art knowledge.

Comment: Figure 4: The shaded area is clearly larger than 1. But the integral from the probability density function over all temperatures should be equal to one. Please, norm the probability values in both figures.

Response: We have specially checked this issue one more time and find that the shaded area is exactly equal to one. For better understanding, we will change the plot type from shaded area to a histogram with vertical bars.

Comment: Figure 7: There should be some warning (or another "name" besides "case") that cases in Fig. 7 are not the same as in Fig. 3, Fig. 5 and Fig. S4.

Response: We agree with Referee that using different cases could create a misunderstanding. We will add according warning to the caption of the figure.

Please also note the supplement to this comment:
https://www.atmos-chem-phys-discuss.net/acp-2018-569/acp-2018-569-AC1-supplement.pdf

---

## Referee Comment (RC2) · Anonymous Referee #3 · 14 Nov 2018

This is a very well-conducted analysis on an urban heat island effect in a high-latitude urban center during winter conditions. The paper is well written and easy to follow. I have a few minor issues to be considered before accepting the paper for publication.

The authors show that roughly up to half of the observed temperature anomaly between the sites U1 and R1 can be due to orographic effects during cold winter days, not due to a real urban heat island (UHI) effect. Yet, the define UHI directly as this temperature anomaly (page 5, lines 28-29). I would very much recommend that the authors call the observed temperature differences between any two (urban vs. rural) sites as temperature anomalies, or something related to that, but not UHI.

Page 5, line 28: Please mention explicitly in the text that $T_i$ and $T_j$ refer to temperatures at sites i and j, respectively. Furthermore, it might be worth mentioning that they are

2-m air temperatures, simply because later in the paper also land surface temperatures are being discussed.

Page 6, lines 11-12: this should rather read "…make many of the proposed UHI scalings…"

---

## Author Comment (AC2) · 19 Nov 2018

We thank the Anonymous Referee for careful reading of our discussion paper. We have found a number of his/her suggestions to be very useful for our study and have revised the manuscript accordingly. A number of other minor stylistic edits have been applied. All of the changes in the revised manuscript are highlighted in green. The revised manuscript is attached. A detailed response to each of the comments is presented below and duplicated in the attached PDF.

––––––––––––––––––––––––––––––––

Comment: The paper presents results of an unique observation dataset, obtained in the Russian polar area, specifically in the Apatity city and its surroundings. Observed

data together with satellite measurements and high-resolution model results give an interesting view in the urban heat island in polar region.

Response: we would like to thank the Referee for so high appreciation of our work, that inspires us for further improvements of our study and further developments of this research direction.

————————————————————

Specific comments:

Comment: Page 2, line 15: The winter UHI in mid-latitude cities is not so environmental problem, but it can be significant and also dependent on the anthropogenic heat release. E.g., Bohnenstengel et al. (2012) conclude impact of AH to 1.5 K in December in London. Also model studies (Trusilova et al., 2016; Huszar et al., 2014) found the winter UHI in central European cities.

Response: We agree with the Referee that the UHI in the mid-latitude cities could be significant and intensive also in winter months. Although the wintertime UHI has been recognized in the studies referred to by the Referee and in our manuscript, there were relatively little attention to it so far. Our study does not object the existence of the wintertime UHI in the mid-latitude cities. But it maintains that the winter UHI is not an important environmental problem in contrast to the summer UHI there. We specifically mention that the mean winter UHI is less intensive than that mean summer UHI as the referred studies have demonstrated.

Changes in the manuscript: no changes regarding to this comment.

————————————————————

Comment: P. 5, l. 23: What does it mean TERRA_URB switched off – removing of urban fraction from model grid-boxes (annihilation approach, e.g. Baklanov, 2016) or only not using of TERRA_URB parametrization? The second option admits that still some physical properties of the surface are altered for urban grid cells (by default in

the model).

Response: The noURB run follows the annihilation approach. The TERRA_URB scheme was switch off AND the urban fraction was removed from model grid cells. The urban fraction in noURB run was set to zero. The land-cover parameters for the urban grid cells were set equal to the nearest non-urban grid cells.

Changes in the manuscript: We have clarified this issue and refer to the annihilation approach in the text (p. 5 l. 23-25). The piece of the text, which was modified regarding to this comment, is presented below: "The model simulations were conducted for the winter season of 2015/16. They included three separate numerical experiments: the URB_AHF run with the TERRA_URB scheme switched on and the AHF defined as described above; URB_noAHF with TERRA_URB switched on and zero AHF and noURB run designed according to annihilation approach (Baklanov et al., 2016), which means that TERRA_URB scheme was switched off, AHF was set to zero and land-cover parameters for the urban grid cells were set equal to the nearest non-urban grid cells."
* * *
Comment: P. 7, l. 10: The conclusion about AHF is too fast. Despite a very low solar radiation, the UHI is not created only by AHF. E.g., a reduced long-wave radiation in the urban environment (due to reduced sky-view factor) can also contribute to the UHI formation in calm anticyclonic situations. Similarly in discussion (p. 8, l. 10-11), the driver could be mentioned.

Response: We agree with the Referee that the AHF scale and impact need further investigation. Concluding the analysis of observations, we formulated a hypothesis about the impact of the AHF on the UHI. We study this hypothesis later in the manuscript, presenting the results of the sensitivity experiments with COSMO-CLM model. In the text, we emphasize that there are other plausible drivers of the UHI in this city. Those drivers, such as long-wave radiation and sky-view factors, have not been addressed in

this study. Nevertheless, the model experiments suggest that there is not a big heat budget imbalance to be attributed to other factors then the AHF.

Changes in the manuscript: no changes regarding to this comment.

––––––––––––––––––––––––––––––

Comment: P. 8, l. 5: It is inaccurately to attribute the temperature difference between U1 and R1 as the "UHI intensity" and moreover, compare it with values for listed megacities. The U1-R1 difference is created not only by anthropogenic factors. Only about 50 % (1 K) is caused by UHI effects, as correctly written in the conclusion.

Response: We would like to highlight that we specifically avoid calling the temperature difference between the U1 and R1 sites as the UHI intensity in our study. However, such pair of stations characterizes the difference between the city and the nearest WMO station. Without specific knowledge about local microclimates, such difference is often associated with UHI intensity. Strictly speaking, we do not compare our data with the numbers given for the biggest megacities. We used those numbers to put the study into a recognized context. We show to important facts in the perceived comparison: 1) there are very few (typically just one) relevant meteorological stations to get climate information for a medium city, such as Apatity, and this station is not representative for urban environment; 2) there is a surprisingly strong urban-rural temperature difference in Apatity, which is a motivation for further investigation of the driving factors of such difference.

Changes in the manuscript: in order to avoid possible misunderstanding, we have revised the usage of the term "UHI intensity" in our study. We have replaced it to "urban temperature anomaly" or to the mathematic designation $T\_R1\hat{}U1$ in all cases when it was specifically related to the temperature difference between U1 and R1 sites, including the paragraph where we introduce such temperature difference (p. 6 l. 1-6). Multiple small changes, related to this comment, are highlighted by green in the revised manuscript, and the modified introduction of $T\_R1\hat{}U1$ is presented below:

"We use a temperature difference T_R1^U1=T_U1-T_R1 to quantify the urban temperature anomaly. Such difference represents the deviation of the air temperature in the city center (U1 site) from the nearest WMO station (R1 site). The WMO station is used as a baseline source of weather- and climate related information in the studied area, so $\Delta$T_R1^U1 represents the deviation of the actual temperature in the city from the regional baseline value. In many UHI studies such temperature differences are associated with UHI intensity. However, R1 and U1 sites are found at different elevations (132 and 180 m above the sea correspondingly) are situated dinfferently with respect to the local orograhy features. Besides, the WMO station is situated close to the Imandra Lake, but our analysis consider only winter conditions with a frozen lake surface, so the influence of the water area can be excluded. Below, we will examine the effects induced by the 48 m elevation difference and orograhy effects as well as by the anthropogenic UHI drivers."

————————————————————

Comment: P. 11, l. 5: There is no clear evidence for the conclusion "The AHF during extreme cold days may warm the city center by up to 6 K". The 50% contribution of all anthropogenic impact is in the time average (as you wrote in p. 9., l. 28).

Response: We do not agree with the Referee. Our observations and modelling results, presented in this study, show that the given estimate of 50% is valid both for the time average and for the cases with the observed extreme urban-rural temperature difference. Moreover, higher anthropogenic contributions could be expected during the coldest days due to increase of heating. We will focus on more accurate and detailed estimates in the further studies.

Changes in the manuscript: we have added the explicit description of our estimates of anthropogenic UHI intensity to the text (p.9, l. 28-31). The piece of the text, which was modified regarding to this comment, is presented below:

"Coherent evidence from the modelling results, in situ and remote sensing data indi-

[Figure]

cate that the anthropogenic contribution to the observed urban temperature anomaly in Apatity ($\Delta T\_R1^{\wedge}U1$) is about 50% (and likely more in the very cold days). Hence, the intensity of the anthropogenic UHI for Apatity could be estimated as 1K for wintertime average with extremes up to 6K. The rest of the observed temperature anomaly is caused by the local orography variations."

———————————————

Comment: 7) Abstract: The values of 1.9 K and 11 K are misleading, because there is no information about different altitudes, which has a significant impact (of the same magnitude as the anthropogenic impact).

Response: The given numbers shows the apparent observed temperature difference. Nowhere in the manuscript this difference is referred to as the UHI intensity. Moreover, we investigated what part of this observed difference could be attributed to the UHI intensity within the limitations imposed by the observation network, methodology, period of study and the limitations of the utilized modeling experiments. This context is clarified further down in the Abstract. At the same time, we do agree that those numbers are hooking a potential reader and call for more extensive research.

Changes in the manuscript: no changes regarding to this comment.

———————————————

Comment: Abstract: The sentence "... direct anthropogenic heating contributes at least 50% to the observed UHI intensity, and the rest is created by natural microclimatic variability..." is wrong, because the UHI intensity cannot be created (in principle) by "natural micro-climatic variability...". The true sentence is "At least 50% of this warm anomaly is caused by the UHI effect, driven mostly by direct anthropogenic heating.", as written in conclusion.

Response: We agree with Referee about this point.

Changes in the manuscript: we have revised the Abstract according to Referee's suggestion (p.1, l. 24-25). The piece of the text, which was modified regarding to this comment, is presented below: The statistical analysis and modelling suggest that at least 50% of this warm anomaly is caused by the UHI effect, driven mostly by direct anthropogenic heating, and the rest is created by natural microclimatic variability over the undulating relief of the area.
* * *
Technical corrections:

Comment: P. 3, l. 15: The sentence and all paragraph (comparison with other studies) belongs rather to discussion. In this part of introduction, aims of study should be specified.

Response: The aim of the study specified above, see p.3 l. 11 in original manuscript (and p. 3 l. 13 in revised). The paragraph, indicated by Referee, refers to the main findings of the study in the context of state-of-the-art knowledge.

Changes in the manuscript: to emphasize the aim of the study, we have made a new paragraph that begins from the corresponding sentence (p. 3 l. 13).
* * *
Comment: Figure 4: The shaded area is clearly larger than 1. But the integral from the probability density function over all temperatures should be equal to one. Please, norm the probability values in both figures.

Response: We have specially checked this issue one more time and find that the shaded area is exactly equal to one.

Changes in the manuscript: For better understanding, we have changed the plot type in Figure 4 from shaded area to a histogram with vertical bars. Additional information was added to the capture of the figure:

Figure 4. The observed daily mean (a) and daily maximum (b) temperature difference

$\Delta$T_R1^U1 (circles) as a function of (a) the daily mean and (b) the daily minimum temperature at the R1 station during the winter of 2015-2016. The red line shows the linear regression. The blue vertical bars show the probability distribution of the (a) daily mean and (b) daily minimum temperature at the R1 station for the winter months (DJF) of 1986-2016. Probability is calculated for temperature bins with 2 K step.
* * *
Comment: Figure 7: There should be some warning (or another "name" besides "case") that cases in Fig. 7 are not the same as in Fig. 3, Fig. 5 and Fig. S4.

Response: We agree with Referee that using different cases could create a misunderstanding. We have added according warning to the caption of the figure.

Changes in the manuscript: we have added a sentence "These periods differ from cases 1-6 that were used before" to the capture of Figure 7.

Please also note the supplement to this comment:
https://www.atmos-chem-phys-discuss.net/acp-2018-569/acp-2018-569-AC2-supplement.zip

---

## Author Comment (AC3) · 19 Nov 2018

We thank the Anonymous Referee for careful reading of our discussion paper. We have find his/her suggestions to be very useful for our study and have revised the manuscript accordingly. A number of other minor stylistic edits have been applied. All of the changes in the revised manuscript are highlighted in green. The revised manuscript is attached.

A detailed response to each of the comments is presented below and duplicated in the attached PDF file.

————————————————

Comment: This is a very well-conducted analysis on an urban heat island effect in a

high-latitude urban center during winter conditions. The paper is well written and easy to follow. I have a few minor issues to be considered before accepting the paper for publication.

Response: we would like to thank the Referee for so high appreciation of our work, that inspires us for further improvements of our study and further developments of this research direction.

—————————————————

Comment: The authors show that roughly up to half of the observed temperature anomaly between the sites U1 and R1 can be due to orographic effects during cold winter days, not due to a real urban heat island (UHI) effect. Yet, the define UHI directly as this temperature anomaly (page 5, lines 28-29). I would very much recommend that the authors call the observed temperature differences between any two (urban vs. rural) sites as temperature anomalies, or something related to that, but not UHI.

Response: We agree with the Referee about this issue. We would like to highlight that we have been originally avoiding calling the temperature difference between the U1 and R1 sites as the UHI intensity in our study, however the related terminology was not very strict. The Referee's comment has motivated us for more accurate revision of the usage the "UHI intensity" term in our study. As he/she suggested, we have replaced the "UHI intensity" to "urban temperature anomaly" or to the mathematic designation $T\_R1^U1$ in all cases when it was specifically related to the temperature difference between U1 and R1 sites, including the paragraph where we introduce such temperature difference (p. 6 l. 1-6).

Changes in the manuscript: multiple small changes, related to this comment, are highlighted by green in the revised manuscript, and the modified introduction of $T\_R1^U1$ is presented below:

Interactive
comment

"We use a temperature difference T_R1ˆU1=T_U1-T_R1 to quantify the urban temperature anomaly. Such difference represents the deviation of the air temperature in the city center (U1 site) from the nearest WMO station (R1 site). The WMO station is used as a baseline source of weather- and climate related information in the studied area, so ∆T_R1ˆU1 represents the deviation of the actual temperature in the city from the regional baseline value. In many UHI studies such temperature differences are associated with UHI intensity. However, R1 and U1 sites are found at different elevations (132 and 180 m above the sea correspondingly) are situated dinfferently with respect to the local orograhy features. Besides, the WMO station is situated close to the Imandra Lake, but our analysis consider only winter conditions with a frozen lake surface, so the influence of the water area can be excluded. Below, we will examine the effects induced by the 48 m elevation difference and orograhy effects as well as by the anthropogenic UHI drivers."

————————————————

Comment: Page 5, line 28: Please mention explicitly in the text that Ti and Tj refer to temperatures at sites i and j, respectively. Furthermore, it might be worth mentioning that they are 2-m air temperatures, simply because later in the paper also land surface temperatures are being discussed.

Response: We agree with the Referee about this issue. Corresponding information has been added (see p. 5 l. 30).

Changes in the manuscript: The piece of the text, which was modified regarding to this comment, is presented below:

"Analysis in this study is based on a temperature difference ∆T_jˆi=T_i-T_j, where T_i and T_j refer to measured air temperature at sites i and j, respectively."

————————————————

Comment: Page 6, lines 11-12: this should rather read "...make many of the proposed
UHI scalings..."

Response & changes in the manuscript: We agree with the Referee about this issue. Corresponding edits have been applied (see p. 6 l. 12).

Please also note the supplement to this comment:
https://www.atmos-chem-phys-discuss.net/acp-2018-569/acp-2018-569-AC3-supplement.zip
* * *